# Hybrid low-voltage physical unclonable function based on inkjet-printed metal-oxide transistors

Alexander Scholz[1,2,6], Lukas Zimmermann [3,6], Ulrich Gengenbach[4], Liane Koker [4], Zehua Chen[4], Horst Hahn [1], Axel Sikora[3], Mehdi B. Tahoori [5] & Jasmin Aghassi-Hagmann [1,2✉]

Modern society is striving for digital connectivity that demands information security. As an emerging technology, printed electronics is a key enabler for novel device types with free form factors, customizability, and the potential for large-area fabrication while being seamlessly integrated into our everyday environment. At present, information security is mainly based on software algorithms that use pseudo random numbers. In this regard, hardware-intrinsic security primitives, such as physical unclonable functions, are very promising to provide inherent security features comparable to biometrical data. Device-specific, random intrinsic variations are exploited to generate unique secure identifiers. Here, we introduce a hybrid physical unclonable function, combining silicon and printed electronics technologies, based on metal oxide thin film devices. Our system exploits the inherent randomness of printed materials due to surface roughness, film morphology and the resulting electrical characteristics. The security primitive provides high intrinsic variation, is non-volatile, scalable and exhibits nearly ideal uniqueness.

[1] Institute of Nanotechnology, Karlsruhe Institute of Technology, Hermann-von-Helmholtz-Platz 1, Eggenstein-Leopoldshafen 76344, Germany. [2] Institute for Applied Research, Offenburg University of Applied Sciences, Badstraße 24, Offenburg 77652, Germany. [3] Institute of Reliable Embedded Systems and Communication Electronics, Offenburg University of Applied Sciences, Badstraße 24, Offenburg 77652, Germany. [4] Institute for Automation and Applied Informatics, Karlsruhe Institute of Technology, Hermann-von-Helmholtz-Platz 1, Eggenstein-Leopoldshafen 76344, Germany. [5] Chair of Dependable Nano Computing, Karlsruhe Institute of Technology, Haid-und-Neu-Straße 7, Karlsruhe 76131, Germany. [6]These authors contributed equally: Alexander Scholz, Lukas Zimmermann. ✉email: jasmin.aghassi@kit.edu

The term Internet of Things (IoT) describes the ubiquitous presence of interconnected devices exchanging sensitive data, often through open communication channels, which can be compromised. In this era of digitization, device identity and encryption techniques are more than ever vital factors for ensuring data security, proper authentication, and secure communication[1–8]. Many security solutions, such as the asymmetric key establishment methods[9,10], are based on mathematically proven foundations and significantly rely on random numbers. The trustworthiness of the encryption depends on the quality of the random number generators (RNGs) used for the generation of public and private keys. Ideally, the generation of random numbers uses the entropy of a high-quality randomness source. Various software-based computational or hardware-based methods exist for RNGs, each more or less satisfying the requirements for cryptographic applications. Fast symmetric cryptographic algorithms, such as the Advanced Encryption Standard (AES)[11], have also been used and offer a high level of security, if the established private key is truly secret. As a consequence, interlinked security solutions in software as well as in hardware are increasingly in demand. Especially for IoT-enabled devices, which are often subject to strict design and performance constraints, hardware-based security is essential.

To tackle the aforementioned challenges, hardware-intrinsic security solutions based on unique device parameters are deployed. These security primitives are referred to as physical unclonable functions (PUFs), which are used for identification, authentication, and cryptographic key generation[12–15]. Among the main advantages of PUFs to be used as root of trust in the IoT, is their unique and unclonable response as well as their memory-less circuit architecture making them less vulnerable to invasive attacks. Historically, initial studies regarding PUFs were carried out by exploiting imperfect surface speckle patterns[16] as an entropy source. One major drawback is the high equipment overhead for key generation and readout, making this PUF approach inapplicable for most application fields. In the next step, silicon-based PUFs (Si-PUFs), leveraging intrinsic manufacturing-induced variations of integrated circuits (ICs), were studied and present a well-established PUF implementation[5,12–14,17,18]. Si-PUFs can be further classified as bi-stable[19–21], delay-based[22], or analog PUFs[23,24], depending on the underlying variation source, such as dopants, defects, and geometrical device dimensions. Furthermore, Si-PUFs often suffer from reduced entropy[25], compared with additive manufacturing techniques. Also, IC fabrication is limited to a few foundries worldwide. In many application domains where the entire design and fabrication are sensitive, globalization of foundries is a major risk. This invites counterfeiting and tampering of the IC or readout of the Si-PUF-specific keys by third parties before the official deployment by the end user. In general, lithography-based silicon technologies face a cost barrier for many envisioned applications and devices in the scope of the IoT. Furthermore, with the ever-increasing demand in electronic devices for the IoT, electronic waste (e-waste), that presents a substantial ecological problem up to now[26–28], is even expected to increase[27,28].

In the near future, emerging technologies such as printed electronics (PE) will further expand the IoT by enabling features like new form factors, flexible substrates, stretchability of the hardware circuits, transparency, and large-area sensing while providing low-cost fabrication[29–31]. PE can help to tackle e-waste by efficient use of non-toxic, and bio-degradable materials to empower a sustainable future regarding "green electronics"[32,33]. Recently, PUFs based on novel materials, such as quantum dots (Liu et al.[34]), biological human T cells (Wali et al.[25]) were presented, mostly utilizing optical inspections for key generation. Other works on promising novel materials and methods

concentrate on fingerprint-alike intrinsic counterfeit protection using random surface patterns[25,34–41]. These approaches need high-cost equipment such as microscopes, image processing, and optical readout for reliable key generation. Also, optical PUFs may face challenges when integrated into signal- and software-layers, which are mainly realized as electronic systems. Another direction in PUF research is to use novel materials and methods and adapt the concepts of Si-PUF architectures[42–50]. Emerging technologies, such as PE, enable decentralized, customizable manufacturing of small and mid-size volumes, which can help to establish "root of trust" in the manufacturing supply chain. Furthermore, owing to the manifold possibilities in design, materials, and substrates it offers a core complexity with many sources of entropy, which can be exploited for hardware-based security. Owing to the additive manufacturing process of printed active devices, larger device variations are expected, compared with lithographically structured devices. The source of variations can be numerous, reaching from the discrete droplet size of the printed ink, the surface roughness of printed layers and the quality of interfaces between layers in general, up to scaling errors in the channel width-to-length ratio due to the manufacturing process and physical layout of a design, to name but a few[51–56].

In this work, we leverage the sources of entropy in printed devices to generate a unique unclonable response function. We develop unclonable, unique PUF security primitives by combining the advantages of PE and Si-based technologies. The security primitives are fabricated through inkjet-printing of the PUF core using electrolyte-gated transistors (EGTs). The surrounding control logic that harvests the intrinsic variations of the printed PUF core is composed of Si-based technology. So far, related printed PUF implementations remain limited to standalone, often single device implementations, neither addressing the system level integration nor including statistical validity. With this work, we overcome the existing limitations with an integrated hybrid PUF implementation for 28-bit, which is scalable to arbitrary key bit width and shows superior security features. We have further performed a statistical analysis based on experimental large-scale characterization for important PUF security metrics, such as uniqueness, bit aliasing, reliability, and bit errors. In addition, the design has the potential in the near future of being fully printed, including the Si-based logic periphery circuits, once inkjet-printed circuit technology evolves further. To the best of our knowledge, designing, fabricating, and embedding a printed PUF core into a system level environment as well as the experimental analysis of PUF security metrics has not been presented before.

## Results

**Full fabrication of hybrid PUF security primitive.** The hybrid (i.e., Si-based control logic and printed PUF core) PUF system and its components are shown in Fig. 1a–d. The printed PUF core, as an intrinsic variation source, is embedded into an addressing and readout environment, as shown in Supplementary Fig. 1. The hybrid PUF evaluation and unique response generation principle is shown in Fig. 1e–h and Supplementary Fig. 2. The PUF core circuit consists of an inkjet-printed inverter array realized with EGTs and resistive indium-tin oxide (ITO) meander structures as load resistors, as shown in Fig. 1b. EGTs operate at low voltages ($\leq 2.0$ V) and show circuit performances ranging from several hundreds of Hz until kHz[57–60]. Studies on EGT variations and device variation modeling have also been studied prior to this work[61,62]. In our hybrid PUF approach, we exploit the implicit random variations caused by the material composition, layer thickness and roughness, as well as interface properties between several printed layers as a source of randomness for hardware security (see Fig. 1d). These variations are reflected in

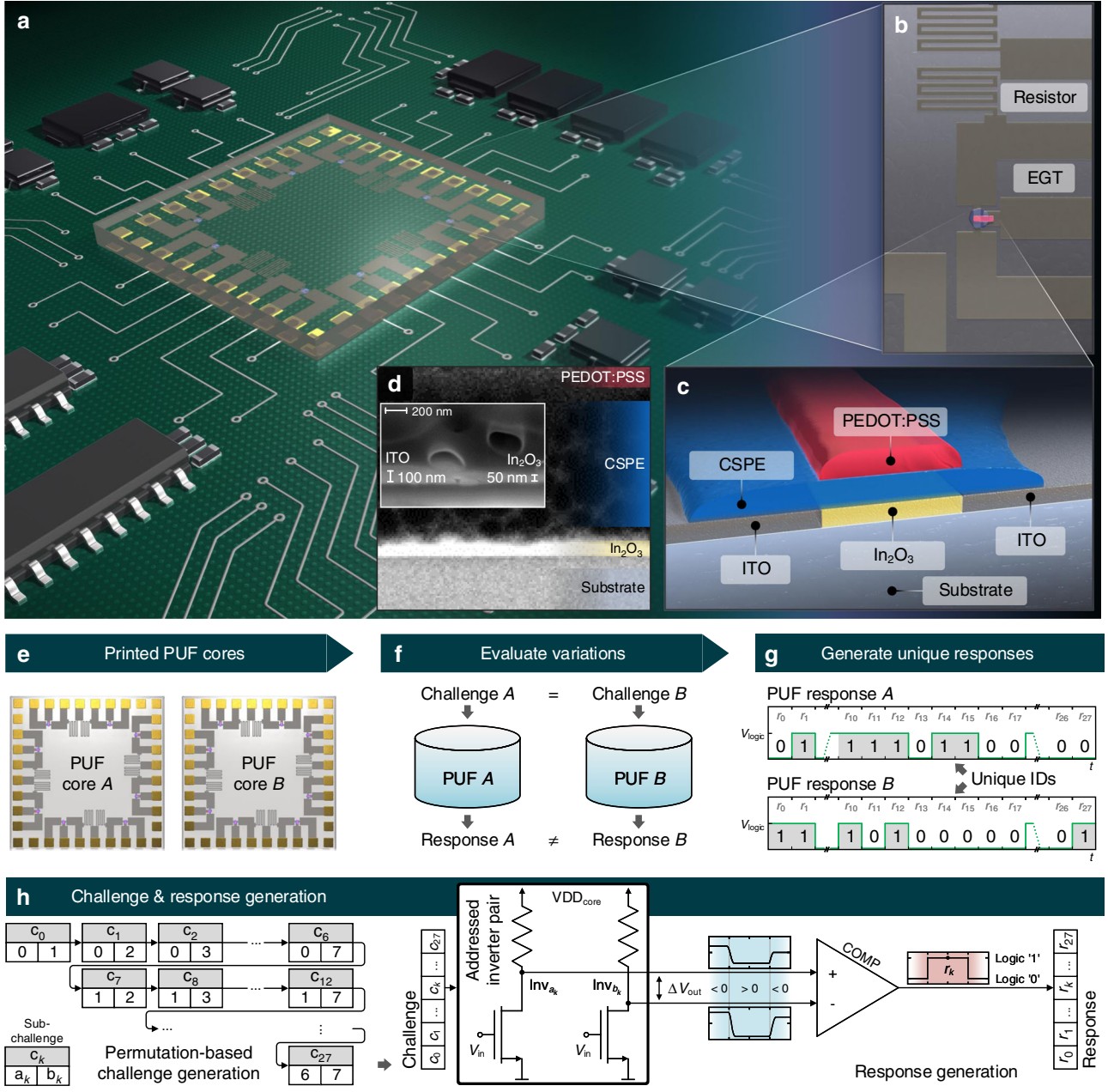

**Fig. 1 Integrated hybrid PUF circuit and corresponding challenge-response mechanism. a** Schematic illustration of the hybrid PUF with the printed core substrate mounted onto the control logic printed circuit board (PCB). **b** Zoom in of one PUF core inverter consisting of a resistor-electrolyte-gated transistor-pair. **c** Schematic EGT top-gate bottom-contact stack structure. **d** Scanning electron microscopy (SEM) image of a printed EGT, which shows the non-uniform semiconductor-/electrolyte interface. The small inlet picture shows the ITO film thickness with the known 100 nm layer thickness as a reference. The inkjet-printed ($In_2O_3$) layer is determined to be $\approx$ 50 nm. On the ITO layer, also an ($In_2O_3$) film can be seen. **e** Two examples of printed PUF cores A and B fabricated with equal processes. **f** Evaluation of the random variations caused by the fabrication process in terms of applying the same challenge to the PUFs A and B and extracting their unique responses. **g** Digital timing diagram representation of the unique PUF responses $\{r_0 r_1 ... r_{27}\}$ for two PUF cores A and B, respectively. **h** High-level schematic of the challenge and response generation procedure. The inverter pair ($Inv_{a_k}$, $Inv_{b_k}$) addressing is provided by the permutation-based sub-challenge $c_k$. The comparator output generates the corresponding sub-response $r_k$, based on the voltage difference $\Delta V_{out}$ between $Inv_{a_k}$ and $Inv_{b_k}$.

the electrical characteristics of our printed EGTs and corresponding inverter structures. By addressing two inverters ($Inv_{a_k}$, $Inv_{b_k}$) simultaneously and comparing their output voltages, one output bit is generated based on the voltage difference $\Delta V_{out}$. To enable a comprehensive understanding of the challenge-response mechanism and the corresponding effects that lead to the response bits, we also track the individual inverter output voltages at the comparator input with an analog-to-digital converter (ADC). The full mechanism of the challenge-response generation is shown in Fig. 1h. The exact challenge configuration to generate a 28-bit response is further described in the Supplementary Fig. 2.

As substrate, we have used ITO-covered 20 mm × 20 mm glass (PGO CEC020S) with a layer thickness of 100 nm. After laser

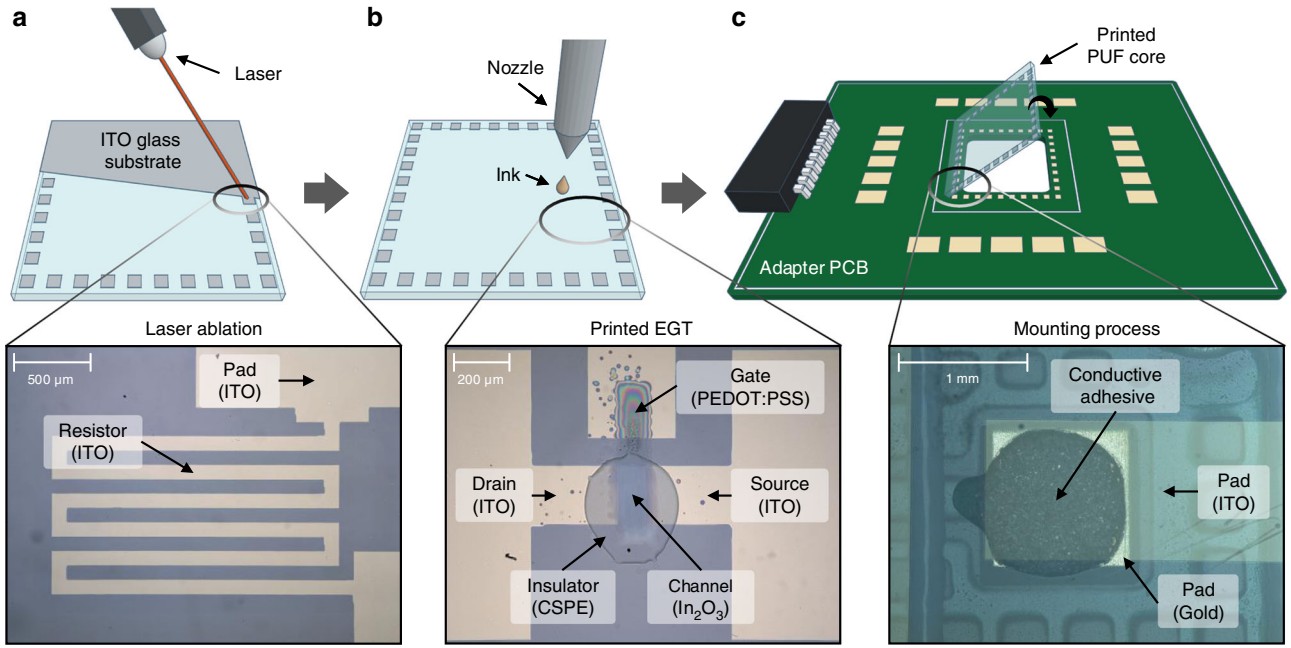

**Fig. 2 Fabrication and integration process of the printed PUF core into the PCB. a** Structuring by laser ablation of the PUF core circuit including routing strips, resistors, I/O and transistor terminals. **b** Inkjet-printing of the EGT layers, such as $In_2O_3$ (channel), electrolyte (gate insulation), and top gate (PEDOT:PSS). **c** Flip-chip adopted mounting process of the inkjet-printed PUF core onto the adapter PCB using conductive adhesive.

ablation, the resulting ITO structures present the electrodes and resistor structures used in the PUF core design, employing eight inverters. In general, the scalable design of the hybrid PUF is not limited to eight PUF core inverters and can be extended if needed. The placement of the components follows a symmetrical alignment, where the input–output (I/O) terminals of all core inverters are connected to bonding pads. This design approach eliminates the need for printed crossovers by shifting this complexity to the Si-based system. A large width of the I/O terminal wiring was chosen to reduce the resistivity of the ITO strips. Also, this increases the manufacturing yield for the printed PUF cores. The EGT transistor stack is designed as a top-gate bottom-contact structure, as displayed in Fig. 1c. Fig. 1d displays the corresponding scanning electron microscopy (SEM) image of a printed EGT, showing the non-uniform semiconductor-/ electrolyte interface as intrinsic variation source. The smaller inlet picture shows the layer thickness as a reference. Fig. 2 shows the manufacturing process and integration workflow for the printed PUF core. In the first step, an ITO glass substrate is structured via laser ablation, using a Trumpf TruMicro 5000 picosecond laser. The structured ITO layer includes the full electrical signal routing of the PUF core such as I/O terminal routing strips, resistive meander structures and the drain, gate, and source terminals of the printed EGTs, respectively.

After cleaning the structured glass substrate in an ultra-sonic bath with acetone and isopropanol (1:1) for 1 hour, an indium (III) nitrate hydrate ($H_2InN_3O_{10}$)-based precursor is inkjet-printed between the transistors' drain and source electrodes. Following an annealing step at 400 °C, the indium-oxide ($In_2O_3$) semiconductor thin-film channel is formed[63], and a composite-solid-polymer-electrolyte (CSPE) is inkjet-printed as a gate insulator. In a final step, poly(3,4-ethylenedioxythiophene):poly (styrenesulfonate) (PEDOT:PSS) is inkjet-printed as the top-gate electrode material. The precursor and ink formulations can be seen in the Methods section. All inkjet-printing steps are performed with a Fujitsu Dimatix DMP-2850 inkjet-printer. The hybrid PUF's control logic includes the addressing and readout circuitry for the printed PUF core and is assembled with

discrete Si-based components (see Supplementary Fig. 1 and Supplementary Note 1). To enable large-scale characterization, an adapter PCB has been designed that allows printed PUF cores with a fixed control logic to be interchanged (see Supplementary Figs. 1 and 8). The integration process of the printed PUF cores onto Si-based adapter PCBs is derived from the silicon flip-chip mounting process[64]. Conductive interconnection between the ITO-based printed PUF core conductive layer I/O terminals and the gold-coated contact pads on the adapter PCB is implemented by a low-temperature curing conductive adhesive. This approach allows mounting at ambient room temperature, thereby diminishing potential negative impact on the printed devices during the integration process. Automated integration includes adhesive dispensing on the adapter PCBs and precise alignment and mounting of the PUF core substrates on the PCBs. The proposed framework allows for various different technology nodes to be seamlessly integrated, in our case printed EGTs and discrete Si-based CMOS circuitry. The standardized interface between printing technology and PCB also permits the evaluation PCB to be interchanged, enabling large-scale automated characterization of several printed PUF cores, for example, for statistical evaluation in a real system environment.

**Printed PUF core analog signal level characteristics.** In general, a full readout of the hybrid PUF includes all PUF core inverter address pair permutations. For eight inverters, the maximum bit-width $L_{max} = 28$ for a single PUF response is calculated by the binomial coefficient without repetitions according to Eq. (1). In the approach used here, we apply all inverter address permutations subsequently in sub-challenges, thereby generating the corresponding sub-response bits, and concatenate them to obtain the full PUF response as a digital bit sequence.

$$L_{max} = \frac{M \cdot (M-1)}{2}\bigg|_{M=8} \quad (1)$$

To obtain more PUF responses, the number of inverter address combinations in the challenge could be decreased, which splits the

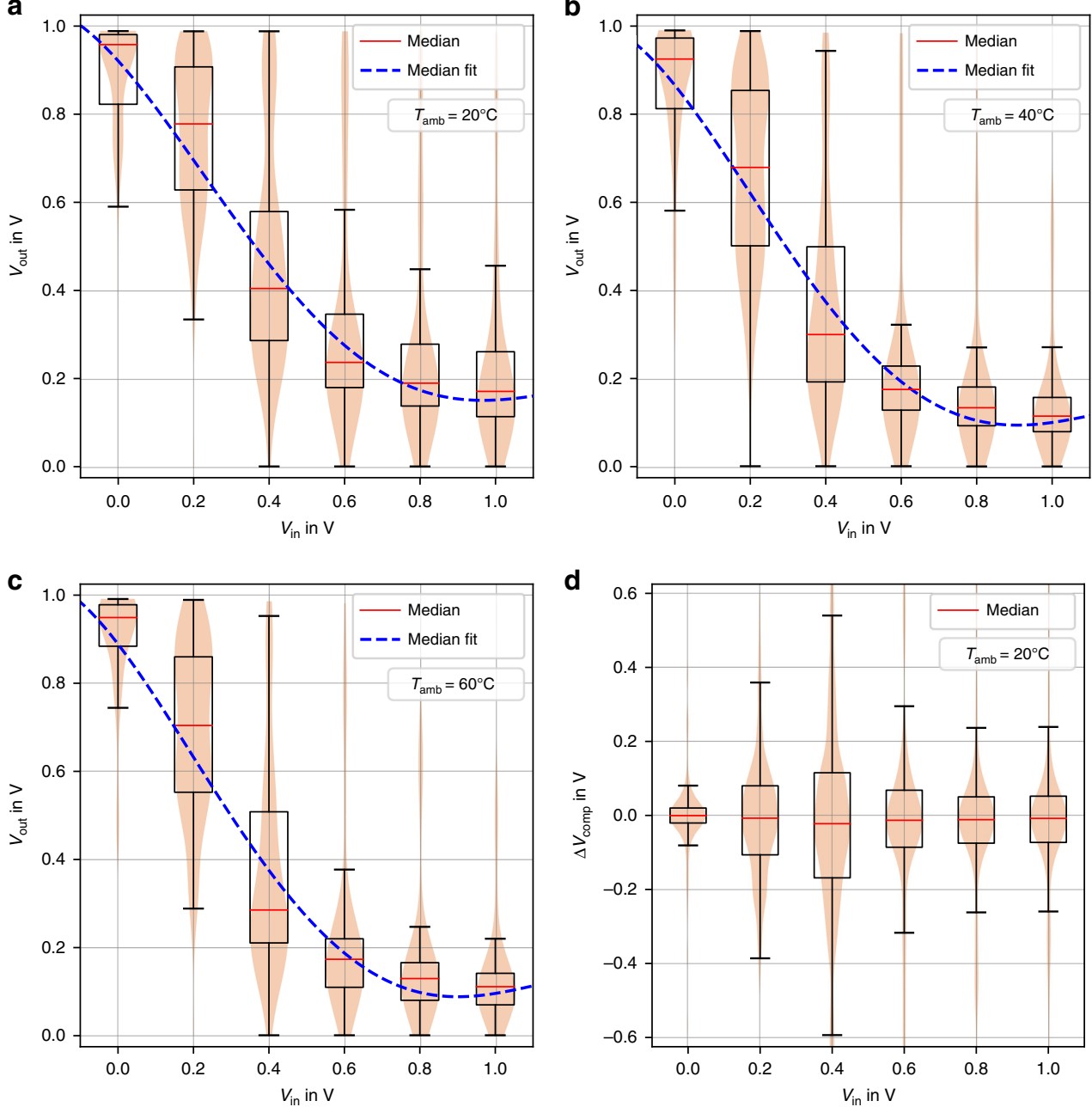

**Fig. 3 Printed PUF core voltage characteristics. a** Inverter voltage transfer curve including all 120 fabricated inverters at a relative humidity of 50%, a PUF core supply voltage of 1 V and at 20 ˚C. The maximum output voltage variation is reached around $V_{in} = 0.4$ V. **b** Inverter voltage transfer curve including all 120 fabricated inverters at a relative humidity of 50%, a PUF core supply voltage of 1 V and at 40 ˚C. **c** Inverter voltage transfer curve including all 120 fabricated inverters at a relative humidity of 50%, a PUF core supply voltage of 1 V and at 60 ˚C. **d** Voltage difference levels $\Delta V_{comp}$ at the comparator input terminals including all PUF core inverter pair combinations at the ambient temperature 20˚C. The data shown in **a**–**d** is visualized as box plot as well as violin plot (light red shaded area). The red line in the box plot marks the median value for $V_{out}$. The upper end of the box in the box plot represents the 75 percentile of the $V_{out}$ distribution and the bottom side the 25 percentile. The maximum is at the 95 percentile, whilst the minimum is at the respective five percentile. The violin plot includes the full data range for $V_{out}$ and helps to visualize the data distribution more clearly. The blue dashed line in **a**–**c** is a fit over the corresponding median values with a fourth-degree polynomial.

challenge-response pair (CRP) space into various units. However, it must be noted that this approach also decreases the bit-width of the single PUF responses. For the following investigations of the fabricated hybrid PUFs on the analog signal level, we apply different test cases in defined by temperatures in the range of $T_{amb} = \{20\,°C, 40\,°C, 60\,°C\}$ as well as controlled relative humidity ($RH$) and PUF core supply voltage conditions ($VDD_{core}$). This allows to visualize the temperature dependencies of the PUF core inverter

voltage transfer curves. All measurements were performed in a Weiss WK3 climatic chamber to obtain a controlled operational environment.

Fig. 3a–c show the voltage transfer curves $V_{out} = f(V_{in})$ of 120 inverters taken from 15 fabricated printed PUF cores as statistical violin and box plot with different temperature conditions, (a) $T_{amb} = 20\,°C$, (b) $T_{amb} = 40\,°C$, and (c) $T_{amb} = 60\,°C$, a stable relative humidity at $RH = 50\%$, and PUF core supply voltage at

$VDD_{core} = 1$ V. For an ambient temperature of 20 °C, the inverters' maximum output voltage variation is at $V_{in} = 0.4$ V, as shown in Fig. 3a. With increasing temperature, an increase in the transistor drain current $I_D$ is expected, which leads to a reduction of $V_{out}$, as can be seen in Fig. 3b and 3c. The light red shaded violin plots at each $V_{in}$ cover the complete $V_{out}$ measurement data. The plots indicate that some inverters do not show the expected voltage transfer curves. Nonetheless, such potential malfunctioning of single inverter cells does not affect PUF operation in a negative manner as long as the PUF responses can be reproduced, which is verified experimentally later in this paper.

Furthermore, Fig. 3d shows the corresponding voltage differences at the comparator input terminals ($\Delta V_{comp}$) over the test case $P_1$ as violin and box plot. At $V_{in} = 0.4$ V the median value for $V_{out}$ is $-48$ mV, and the measured samples interquartile range is at 284.1 mV, respectively. We can conclude that biasing the hybrid PUF around this $V_{in}$ is beneficial, as most probably a high-voltage difference $\Delta V_{comp}$ can be expected. This satisfies the constraint that the inverter voltage difference levels $\Delta V_{out}$ are much larger than the control logic's systematic error $\Delta V_\epsilon$ ($\Delta V_{out} \gg \Delta V_\epsilon$)[65].

**Security metrics**. We evaluate the security metrics of our fabricated hybrid PUFs by using the uniqueness, bit aliasing, and reliability metrics proposed by Maiti et al.[66]. To investigate the uniqueness and reliability metrics further, we evaluate the bit errors in the PUF responses that occur over time. Detailed definitions of these security metrics can be found in the section Methods. In the following discussion, we apply various test cases for evaluation. These test cases compromise various temperatures with $T_{amb} = \{20$ °C, 40 °C, 60 °C$\}$, relative humidity conditions $RH = \{45\%, 50\%, 55\%\}$ and different PUF core supply voltages $VDD_{core} = \{0.9$ V, 1.0 V, 1.1 V$\}$. A tabular overview of all resulting test case combinations can be found in the Supplementary Table 1. We consider changes in $RH$ in our evaluations, because it impacts the functionality of the EGTs[67]. In the following, nominal conditions are defined at an ambient temperature of $T_{amb} = 20$ °C, relative humidity $RH = 50\%$, and $VDD_{core} = 1.0$ V, and used as a reference in security metrics evaluation. All measurements are repeated 20 times under controlled environmental conditions.

*Uniqueness.* The uniqueness metric shows the ability to differentiate PUF instances from each other. When applying the same challenge to different PUF instances, all responses are expected to be unique. The ideal value for the uniqueness metric is 50%, which means that all PUF responses are distinguishable. For our evaluation, we use 15 fabricated hybrid PUF core instances and operate them under nominal conditions. The inverter input-biasing voltage is set to $V_{in} = 0.4$ V, as explored in our prior simulations, to be the best operating point for our PUF design[65]. The experimental results, together with the outcome of the simulations, are shown in Fig. 4a. The mean value and standard deviation of the uniqueness for the measured responses are $\mu_m = 51.1\%$ and $\sigma_m = 15.5\%$, respectively. These values are in good agreement with both the simulation and the theoretical ideal value of the uniqueness metric, which is 50% and means that the PUF entities can be distinguished.

*Bit aliasing.* Bit aliasing (BA) will occur if various PUF instances produce the same response when stimulated with the same challenge. In this case, PUF authentication results in false positives, which degrades the operational capabilities of a PUF for use in applications such as secure device identification. In general, bit aliasing is caused by internal biases, which lead to fixed bits that in the worst case never change their binary values. In general, the bit aliasing metric is associated with the uniqueness metric, as internal bitwise biases lead to multiple appearances of PUF responses for different challenges. As a result, the uniqueness of the CRPs degrades. To evaluate the bit aliasing metric, we utilize the same CRP data obtained from our uniqueness tests under nominal conditions. The inverter input-biasing voltage remains at the theoretically best operating point achieved at $V_{in} = 0.4$ V[65]. Fig. 4b shows the distribution of the calculated bit aliasing values obtained separately for each response bit position. The mean value of the bit aliasing calculated using the measured PUF responses is $\mu_m = 44.5\%$ with a standard deviation of $\sigma_m = 9.3\%$. The plot also proves the statistical coverage with the simulation results obtained in our prior work[65]. The experimentally obtained mean bit aliasing value of $\mu_m = 44.5\%$ indicates a bias of the PUF responses towards logic '0'. However, the simulated theoretical bit aliasing for the presented hybrid PUF is $\mu_s = 49.8\%$. This shows that the bit aliasing can be improved to provide a close to true random bit sequence, suitable for cryptographic applications with the presented approach. The hybrid PUF's response entropy and its capabilities regarding identification is shown in Supplementary Fig. 4 and Supplementary Note 1.

*Bit errors and reliability.* To verify the robustness of our fabricated PUFs regarding their response stability, extensive testing under various controlled environmental conditions in a climatic chamber is executed. Standard reliability measures consider PUF responses generated at certain operating conditions and calculate their bitwise deviations to a fixed reference response, extracted under nominal conditions by utilizing the hamming distance (HD). The HD of two binary strings of equal length is the number of positions with different bit values. This evaluation procedure assumes that the response bits remain stable, if the operational conditions, such as ambient temperature, relative humidity, and supply voltage do not change over time/iterations. Therefore, noise-inflicted, time-dependent operational effects such as flipping bits are not further evaluated. However, this is not the case for circuits, whose internal characteristics might change slightly during operation because of non-ideal devices and other factors, such as aging. We measure in the ambient temperature range $T_{amb} = \{20$ °C, 40 °C, 60 °C$\}$, fixed relative humidity $RH = 50\%$, and inverter supply voltage $VDD_{core} = 1.0$ V. For our evaluations, we increase the inverter input-biasing voltage in the range of $V_{in} = \{0.0$ V, . . . , 1.0 V$\}$ in 0.2 V steps. In Fig. 4c, the markers show the calculated mean bit error values. The plot shows that the bit errors strongly increase between $V_{in} = (0.8$ V, 1.0 V$)$. This effect appears because the PUF core inverters have already reached their logical low levels, where variations are small. Nonetheless, a single flipped bit in a 28-bit PUF response has a relative impact of $\approx 3.6\%$. As the measured relative bit errors of the 28-bit responses are $\leq 2\%$, in average there is less than one bit flip per response. In the following the reliability metric is discussed. In total, we operate four fabricated hybrid PUFs in a climatic chamber under all previously defined operating conditions. The inverter input-biasing voltage is consecutively set to the values $V_{in} = \{0.3$ V, 0.4 V, 0.5 V$\}$, in order to investigate the robustness around the best operating point determined in our prior simulations. We repeat the response extraction 20 times for each test case to achieve expressive statistics. Fig. 4d shows the reliability values for each of the tested PUF cores. The thick black line represents a second-degree regression model fit of the calculated mean reliability values that reaches its maximum of 78.5% at $V_{in} = 0.4$ V. It should be noted that no

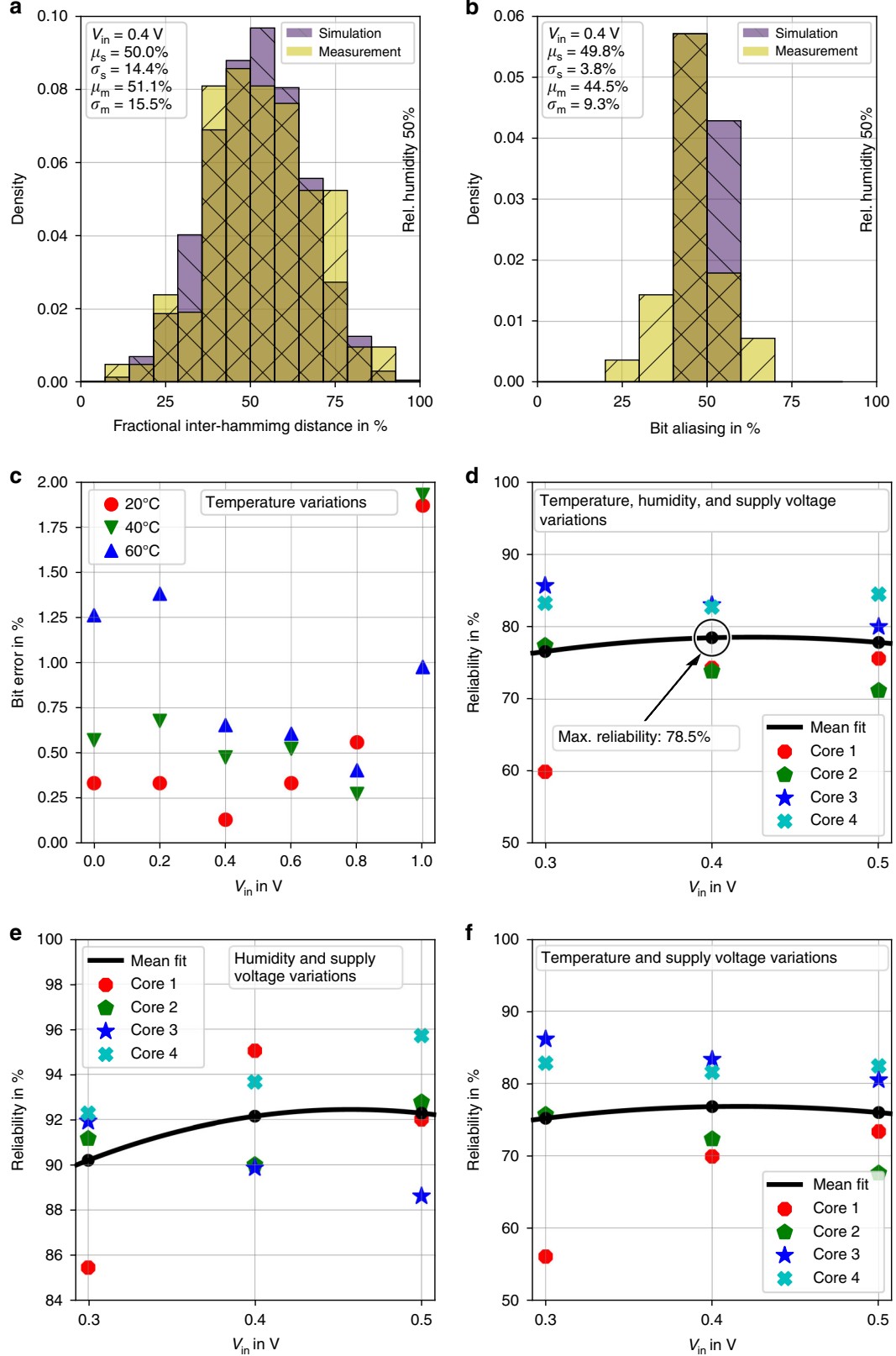

passivation layer for active component protection from environmental impacts was available at the time of fabrication. We expect the reliability values to be higher, once proper passivation is introduced for EGTs. To further investigate the reliability of our fabricated devices without the impact of changing ambient temperatures, we create the plot shown in Fig. 4e. The mean reliability is above 90%, which indicates that values close to ideal can be achieved for our hybrid PUF. Furthermore, the impacts of altered relative humidity and inverter supply voltage VDD$_{core}$ in the range of ±10% are low, respectively.

**Fig. 4 Security metrics for the hybrid PUF. a** Uniqueness at $V_{in} = 0.4$ V, visualized as histogram over the fractional inter-hamming distance for measurement and simulation data. The mean value of the simulated uniqueness is $\mu_s = 50\%$ with a standard deviation of $\sigma_s = 14.4\%$. The measured uniqueness mean value is $\mu_m = 51.1\%$ with a standard deviation of $\sigma_m = 15.5\%$. **b** Bit aliasing at $V_{in} = 0.4$ V, visualized as histogram for measurement and simulation data. The mean value of the simulated bit aliasing is $\mu_s = 49.8\%$ with a standard deviation of $\sigma_s = 3.8\%$. The measured mean value of the bit aliasing is $\mu_m = 44.5\%$ with a standard deviation of $\sigma_m = 9.3\%$. **c** Bit errors for various ambient temperatures over $V_{in} = \{0$ V, $0.2$ V, . . . , $1.0$ V$\}$ and test cases including temperature variations. Each data point displays the mean bit error value of 15 measurement samples. **d** Reliability metric including temperature, humidity, and supply voltage variations. **e** Reliability metric including humidity and voltage variations. **f** Reliability including temperature and supply voltage variations. **d–f** The mean value curve fit is done with a second-degree polynomial.

Fig. 4f shows that the reliability values are not heavily influenced by the tested relative humidity variations. This plot shows the reliability values for the four tested PUF cores with fixed relative humidity at 50%. The data points and curves are almost equal to Fig. 4d.

## Discussion

In conclusion, we have presented a hybrid PUF incorporating an inkjet-printed core circuit as an intrinsic source of entropy, integrated into a silicon-based CMOS system environment. The embedded system can generate 28-bit PUF responses and enables large-scale characterization as well as error tracking on both the digital and analog signal levels. In contrast to other PE-based PUF approaches, the hybrid PUF design presented here covers the full system level integration and breaks through the limitations of former device level-only implementations. Potential target applications in the security domain as well as benchmarking security metrics span the full range from conventional electronics to emerging technology fields, as PE, which enables low-cost manufacturing. Digitally controlled additive and solution-processed manufacturing techniques provide a high degree of flexibility in materials and unique intrinsic variation properties beneficial for fabricating PUF circuits. Flexible substrates in the future also enable new design patterns and integration capabilities, which pave the way for favorable applications in the IoT.

To investigate the security metrics of the hybrid PUF system and to assess the performance of our fabricated hybrid PUFs, we operated and characterized them in the controlled environment of a climatic chamber. In detail, we tested the hybrid PUFs under the impact of temperature, humidity, and supply voltage variations. With a mean uniqueness value of 51.1% and a mean bit aliasing value of 44.5%, the proposed hybrid PUFs generate unique responses that are distinguishable. The bit error evaluation shows that the PUF responses remain stable when regenerated successively at various ambient temperatures, with bit errors ≤2%. To show the robustness around the best operating point obtained from simulations in our prior work, we have shown reliability evaluation for the inverter input-biasing voltages $V_{in} = \{0.3$ V, $0.4$ V, $0.5$ V$\}$. The mean reliability value at the best operating point is at 78.5%. However, the reliability score can be expected to be higher once passivation to protect the EGTs from environmental influences has been introduced. Even with this figure of merit, it is still a major advance compared with the state-of-the-art research. We also investigated the reproducibility of the PUF challenge-response mechanism by calculating the bit errors that occur over time of operation.

Compared with other printed PUFs, our current system shows the largest PUF response bit-width and sufficient statistics to experimentally evaluate security metrics based on characterization results. The hybrid PUF presented here as a scalable, fully integrated system is a promising approach for future utilization in emerging security-related application fields such as identification, multi-factor authentication, and cryptographic key

generation. In addition, inkjet-printing technology enables a root of trust to be established in the PUF supply chain through decentralized manufacturing, which implies a substantial progress in providing trustworthy security primitives. Regarding future work in the area of printed PUFs several points need to be further investigated. As shown in our results, temperature stability needs to be improved to increase reliability and to enable bit-stable PUF responses.

## Methods

**Materials**. As a printing substrate, a commercially available 20 mm × 20 mm ITO glass (PGO CEC020S) with a layer thickness of 100 nm and a sheet resistance of $R_\square = 20\ \Omega$ is used. The materials used for the EGTs are as follows: Indium (III) nitrate hydrate (99.9% trace metal basis, MW = 300.83 , Sigma-Aldrich), Glycerol (MW = 92.09 g mol$^{-1}$, Merck KGaA), Dimethyl sulfoxide anhydrous (DMSO, 99.9%, MW = 78.13 g mol$^{-1}$, Sigma-Aldrich) Propylene carbonate anhydrous (PC, 99.7%, MW = 102.09 g mol$^{-1}$, Sigma-Aldrich), Poly(vinyl alcohol) hydrolysed (PVA, 98%, Sigma-Aldrich), Lithium perchlorate (LP, 99.99% trace metal basis, MW = 106.39 g mol$^{-1}$, Sigma-Aldrich), Poly(3,4-ethylenedioxythiophene) polystyrene sulfonate (PEDOT:PSS, 3.0–4.0% in H2O, Sigma-Aldrich), Ethylene glycol anhydrous (99%, MW = 62.07 g mol$^{-1}$, Sigma-Aldrich). The conductive adhesive, used to contact the core substrates to the adapter PCB is the two-component silver-filled adhesive Elecolit 325 by Panacol.

**Substrate preparation**. The substrates are structured via laser ablation with a Trumpf TruMicro 5000 picosecond laser with a laser wavelength of 1030 nm, set at 2.5 W average power.

**PUF fabrication**. For all printing operations, print heads of the type Fujitsu Dimatix DMC-11610 with an average droplet volume of 10 pl are used. The annealing of the substrate for the indium channel formation is done in a box furnace. The temperature is ramped up to 400 °C for 2 hours and is kept at this temperature level for further 2 hours. For cooling, the substrate is kept in the box furnace over night. It should be mentioned, that other possibilities for channel formation can be utilized, such as photonic or chemical curing, to lower processing temperature[68,69].

**Ink preparation**. The inkjet-printable precursor ink for the indium-oxide channel is based on dissolved 0.05M ($H_2InN_3O_{10}$) in double-deionized water and glycerol with a ratio of 4:1[67]. The solution is stirred for 2 hours and filtered with a 0.2 µm polyvinylidene fluoride (PVDF) filter before usage. The CSPE is prepared with 1 wt % LiClO$_4$, dissolved in 9 wt% PC and stirred for 2 h at ambient room temperature. In addition, 4.29 wt% PVA is dissolved in 85.71 wt% DMSO and stirred at 90 °C for 2 hours. The two solvents are mixed together and stirred until a clear solution is obtained[67]. Before inkjet-printing, the solution is filtered using a 0.2 µm polytetrafluoroethylene filter. In the last step, 70 wt% PEDOT:PSS is dissolved in 30 wt % ethylene glycol and stirred for 2 hours, then filtered with a 0.2 µm PVDF membrane before printing.

**Mounting inkjet-printed PUF core substrate onto adapter PCB**. A mounting process derived from flip-chip-technology has been selected for integrating the inkjet-printed PUF core onto the Si-based adapter PCB. The I/O terminals of the PUF core have to face the contact pads of the adapter PCB. The electrically conductive interconnection is realized with conductive adhesive. The integration process is based upon a mounting process for surface-mounted devices on low-temperature flexible substrates[70] that is adapted to the present application. The four axis cartesian handling platform applied for the mounting process is equipped with an application-specific carrier for the 20 mm × 20 mm PUF core glass substrates, a top camera to identify the fiducials on the PCB, a bottom camera to capture the fiducials on the PUF core glass substrates, a vacuum gripper, and a pressure time-controlled dispenser tool that is adapted to apply small amounts of conductive adhesive.

The silver-filled two-component adhesive Elecolit 325 from Panacol is applied as a conductive adhesive. A small amount of adhesive of 1 g per session is mixed

manually. In the present application, it has a pot time of ≈90 minutes and is used to contact four PUF cores with 28 I/O terminals each to the respective adapter PCBs. The area of the ITO-based conductive layer I/O terminals as well as the gold-coated contact pads on the adapter PCB are 1 mm × 1 mm, respectively. This dimensioning allows the positioning tolerances of the dispenser needle, the PCB and the PUF core glass substrates to be compensated. As examples, 56 contact points were evaluated based upon optical acquisition and image processing, showing that the diameters of the conductive adhesive in the final setup with the assembled glass substrate vary between ≈570 µm and 970 µm.

The integration process starts by manually placing the glass substrates with the printed PUF cores and one adapter PCB in the handling platform. The machine is initialised and an automated image processing-based routine is used to identify the exact position of the PCB. The conductive adhesive is then filled into the dispenser and the position of the dispenser needle is also identified by image processing. After a pre-dispensing step executed as a precondition for enabling reliable and reproducible adhesive flow, the adhesive is automatically dispensed onto the PCB contact pads. The first PUF core is automatically gripped and orientated into the correct mounting position. In order to orientate the glass substrate, the fiducials of the PUF core are acquired by the lower camera and identified based upon an image processing routine. After the glass PUF core has been mounted, the assembled device is manually removed and the next PCB is placed in the machine. The mounting process is then rerun with the next PUF core. Assembling one PUF core on the adapter PCB takes ~10–15 minutes. Final mechanical stability and electrical conductivity of the interconnection is achieved after 16 hours when curing at room temperature. Curing can be accelerated at elevated temperatures.

**Measurement setup and characterization.** A computer-driven automatic challenge-response readout and measurement system was custom-built for hybrid PUF characterization. The hybrid PUF platform design was split into the three functional units (a) microcontroller development board, (b) control logic, and (c) PUF core adapter, as further explained in Supplementary Fig. 8. and Supplementary Note 1. This setup enables the printed PUF core instances to be interchanged, as well as large-scale characterization. The custom-built software implementation allows challenges to the PUF to be applied automatically and the internal analog signals measured with a 12-bit MAX1237EUA ADC that lead to response bit generation. The microcontroller development board is connected to a personal computer (PC) and powered via USB. The control logic's components are powered using the microcontroller development board's internal 5 V and 3.3 V supply pins. For dynamic voltage generation of $V_{in}$ and $VDD_{core}$, two internal 12-bit digital-to-analog converter channels of the microcontroller board are utilized. All our PUF response measurements were performed in a Weiss WK3 climatic chamber under controlled ambient temperature and humidity conditions. For PUF bit error and reliability measures, the ambient temperature was divided into three sections in the range of 20 ˚C to 60 ˚C. The relative humidity was varied in three sections from 45% and 55%.

**PUF security metrics**

*Uniqueness.* The uniqueness metric measures the correlation of the PUF responses from different instances of the same type, by applying the same challenge. The lower the correlation, the greater the uniqueness. Ideally, all PUF responses should differ due to the random intrinsic variations, which implies a uniqueness value of 50%. For two different PUF instances $i$ and $j$, each with $L$-bit responses $R_i$ and $R_j$, the uniqueness for $N$ PUFs in total is defined as:

$$U = \frac{2}{N(N-1)} \sum_{i=1}^{N-1} \sum_{j=i+1}^{N} \frac{HD(R_i, R_j)}{L} \times 100\% \qquad (2)$$

*Bit aliasing.* The bit aliasing metric is a measure of the 0's and 1's distribution across different PUF entities of the same type. Ideally, both binary values occur with the same probability of 50%. If the $l$-th bit of the tested PUF responses has the same bit value across all PUF entities, the inter-HD of this bit will be zero. As a result, various PUFs may produce the same responses, which degrades the uniqueness and leads to false positives in device authentication. The bit aliasing for $N$ PUF entities at the $l$-th bit position is calculated as:

$$BA = \frac{1}{N} \sum_{n=1}^{N} R_{n,l} \times 100\% \qquad (3)$$

*Bit errors.* Bit errors are bit flips that may occur over time when generating the same PUF response multiple times. Owing to changing operating conditions, the regenerated challenge-response pairs might differ, which degrades both the uniqueness and the reliability scores. We calculate the bit errors ($BE_n$) for PUF instance $n$ by using the $L$-bit reference response $R_{ref,n}$ at nominal conditions, and the $L$-bit response $R'_{n,w}$ regenerated $W$-times:

$$BE_n = \frac{1}{W} \sum_{w=1}^{W} \frac{HD(R_{ref,n}, R'_{n,w})}{L} \times 100\% \qquad (4)$$

*Reliability.* The reliability metric is a measure of the stability of PUF responses under the impact of varying operating conditions when applying the same challenge. The ideal value is 100%, which means that PUF responses are not affected by any environmental impacts such as noise, temperature, humidity, or unstable

voltage supply. The reliability $REL_n$ for PUF instance $n$ is calculated by using the $L$-bit reference response $R_{ref,n}$ measured at nominal conditions, and the $L$-bit test response $R'_{n,t}$ for $T$ different operating conditions:

$$REL_n = 100\% - \frac{1}{T} \sum_{t=1}^{T} \frac{HD(R_{ref,n}, R'_{n,t})}{L} \times 100\% \qquad (5)$$

## Data availability
The data sets generated and/or analyzed during the current study are available from the corresponding author on reasonable request.

## Code availability
The codes used for data analysis are available from the corresponding author on reasonable request.

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

## Acknowledgements

This work was supported by the Ministry of Science, Research, and Arts of the state of Baden-Wuerttemberg, Germany through the Modellierung, Entwurf, Realisierung und Automatisierung von gedruckter Elektronik und ihren Materialien (MERAGEM) Doctoral Program. J.A.-H. acknowledges support by the German Research Foundation (DFG) under Germany´s Excellence Strategy via the Excellence Cluster 3D Matter Made to Order (EXC-2082/1 – 390761711). Special thanks go to Dr. Torsten Scherer and Vanessa Wollersen of the Karlsruhe Nano Micro Facility (KNMF) for providing the SEM images of our printed transistors (proposal ID: 2019-023028029).

## Author contributions

A.Sc., L.Z., A.Si., M.B.T., and J.A.-H., conceived the project. A.Sc. designed the circuits and fabricated the inkjet-printed devices. L.Z. designed the printed circuit boards hosting the control logic and implemented the software for automated large-scale characterization and security metric evaluation. U.G., L.K., and Z.C. developed the mounting process to integrate the printed into the silicon-based electronics. A.Sc. and L.Z. performed the electrical experiments. A.Sc., L.Z., and J.A.-H. analyzed the simulations and experimental results. J.A.-H. supervised the project. All authors have participated in analyzing the results and contributed to writing the manuscript.

## Funding

## Competing interests

The authors declare no competing interests.
