## [Peer Review File · Nature Communications]

REVIEWERS' COMMENTS

Reviewer #1 (Remarks to the Author):

Review of "Hybrid low-voltage physical unclonable function based on inkjet-printed metal-oxide transistors" by Alexander Scholz et al

Manuscript#: NCOMMS-20-16112-T

This manuscript is well put together at a level of quality suitable for Nat. Comms.

It concerns the realization of physically uncloneable functions (PUFs) using inkjet printing. The use of PUFs is a hot area, especially for IoT. Therefore this paper is potentially a game changer because low-cost inkjet printing is ideal for the level mass proliferation and indispensability required for IoT at large scale.

The paper may be published after the following major revisions:

1. An important alternative motivation for inkjet printing is missing from the introductory discussion. I suggest the authors do a literature search on "green electronics" and cite some key paper. The question of electronic waste is an increasingly important environmental concern and proliferation of IoT devices adds to this issue. The ability to realise circuitry using inkjet printing is in the right direction for addressing sustainability.
2. In the paper there is an emphasis on the random number generator for the motivation. However, this misses other important motivations. The authors can directly motivate PUF importance considering it as an inseparable root-of-trust in the IoT era.
3. Currently, from the way the paper is written, I cannot clearly see how the response is produced. The authors suggest to provide a high-level figure to assist the understanding of this. This is important and should be provided early on in the paper. (1) How the digital response is produced, (2) a clear statement of which variations this PUF utilizes, and (3) what characteristics this PUF exploits, all need to be CLEARLY stated near the beginning.
4. As there are other works related to printed PUF in literature, the authors should provide a qualitative comparison among them (if quantitative comparison is non-trivial). This is to clearly distinguish this work from the others. At moment the clarity here needs improvement.
5. Paper(s) on PUFs were recently published in Nature Electronics (within the last two months). Please do a search and cite these in the introduction to assist motivating the importance and directions in the field.
6. The math is wrongly formatted. Only single letter variables should be in italic. Other items such as: max, amp, RH, comp, out, core, in, BE, BA, HD...etc should have italic removed. If for example you put RH in italic it means $R \times H$. Use this visualisation to correct all; the math.
7. Whilst it is correct that there is always a space between a number and the physical units, the deg Centigrade symbol is an exception. Remove the space here.
8. Please carefully check the references. There are many errors. For example conventions concerning capitalisation are wrong. All journal names should be in title case, not sentence case. All acronyms should be capitalised.
9. On page 2 you have "Si-PUFs can be further classified..." In English, it is verboten to begin a sentence with an acronym. All sentences must begin with a real English word.

Reviewer #2 (Remarks to the Author):

This is a very well written paper with very interesting and well documented results in terms of creating a printed PUF circuit that can serve as a unique intrinsic identifier. The methods and the results are described in great detail that would allow another to replicate the experiment. The material is a definite interest to the research community.

There are several areas where this paper can be improved:

1. page 1 line 15 "high entropy". The paper claims a 28 bit PUF. But there are 8 inverters. The produced bits have 0.45 bias with similar bits on other devices. I guess high entropy is a relative term. You might want to elaborate.
2. Page 1 lines 21 to 29: you are not distinguishing between the true source of randomness vs. something that uses that to produce pseudorandomness (e.g., a cryptographic algorithm). PUF you are describing is in the former category. The "28 bit" PUF does not provide sufficient entropy into the cryptographic algorithms given that there are 8 inverters and 0.45 bias across devices.
3. Page 1 line 32. "PUFs used as hardware security keys". PUFs can be used to derive the keys. A function is not a key. Also the 22% error in your PUF would preclude getting a bit stable key unless lots of post processing is done (and that would be difficult with printed electronics).
4. Page 1 lines 28-29. Loftstrom paper (24) is not a PUF. It does not define its circuit as having an input and therefore is not a function. PUFs came after Gassend 2002 paper which requires input (challenge) and an output with output determined by manufacturing variation. Butterfly PUF (22) is a bistable PUF, not delay PUF. The first delay PUF paper is arguably the Gassend CCS 2002 paper.
5. Page 4 - 5 lines 122 to 127. It is unclear how the 8 inverters are expanded to 28-bit challenge using permutations. Might want to describe that more. Also elaborate whether the bits generated is true entropy or pseudo-entropy.
6. Page 6 line 177. It seems like μ_m of 44.5% represent a min entropy reduction of $-\log(.55)$ reducing 8 bit entropy to 6.8 bits. This assumes adversary can use other like devices to break another via across device bit correlation.
7. Page 8 line 226. A reliability of 78.5%, meaning error rate $> 20\%$ is actually quite high in the silicon PUF world. For identification, this would require an even longer response length for reliable authentication (for a given false positive and false negative goal).
8. Page 8 line 234 "root of trust" usually refers to a bit-stable key. An error rate of 20%+ is difficult to address even with error correction. And on a printed circuit I am not sure how that's going to be addressed. There is a lot of future work here.
9. Page 10 lines 316 to 326. Why use w and t , two different variables. These two eqns looks almost the same. Isn't one just 100 minus the other?
10. Page 11 line 360. I don't see an italicized publication name.

Point-by-point response letter to the reviewers:

We would like to thank the reviewers for their valuable and very positive feedback on our manuscript. We are glad that you find our manuscript “very well written”, the material of “definite interest to the research community” and our work in total a “potentially a game changer”.

We have answered all comments and questions and herewith resubmit our substantially revised manuscript accompanied by a point-by-point response letter. Please find our detailed comments and changes in the manuscript below.

Reviewer #1:

This manuscript is well put together at a level of quality suitable for Nat. Comms.

It concerns the realization of physically uncloneable functions (PUFs) using inkjet printing. The use of PUFs is a hot area, especially for IoT. Therefore this paper is potentially a game changer because low-cost inkjet printing is ideal for the level mass proliferation and indispensability required for IoT at large scale.

Comment 1.1:

An important alternative motivation for inkjet printing is missing from the introductory discussion. I suggest the authors do a literature search on “green electronics” and cite some key paper. The question of electronic waste is an increasingly important environmental concern and proliferation of IoT devices adds to this issue. The ability to realise circuitry using inkjet printing is in the right direction for addressing sustainability.

Response 1.1:

Thank you for pointing this out. Ecological damage dealt by electronic waste (e-waste) is an extremely important issue in these days.

We added this argument to our motivation for inkjet printing of PUFs regarding its potential for empowering “green electronics”. Inkjet-based additive manufacturing allows for minimal material waste incorporating also non-toxic, and ideally bio-degradable materials and solvents. Furthermore, with the expected economic growth pushed by the IoT, a massive increase in deployed electronics is expected. Inkjet-printed electronics can therefore help to counteract ecological damage dealt by electronic waste.

Changes to the manuscript:

Line 44-45, new (added): Furthermore, with the ever-increasing demand in electronic devices for the IoT, electronic waste (e-waste), that presents a substantial ecological problem up to now [26–28], is even expected to increase [27,28].

Line 48-49, new (added): PE can help to tackle e-waste by efficient use of non-toxic, and bio-degradable materials to empower a sustainable future regarding 'green electronics' [32,33].

Comment 1.2:

In the paper there is an emphasis on the random number generator for the motivation. However, this misses other important motivations. The authors can directly motivate PUF importance considering it as an inseparable root-of-trust in the IoT era.

Response 1.2:

We fully agree and have further highlighted the importance of PUFs regarding root of trust in the IoT area accordingly. In order to, in parallel, address a comment from reviewer 2, we changed the sentence in line 31-32. Therefore, the motivation for PUFs as root of trust in the IoT is strengthened in the introduction which also addresses comment 2.3 of reviewer 2, since the word “hardware security keys” is replaced by “root of trust”.

Changes to the manuscript:

Line 31-32, old: Among the main advantages of PUFs to be used as hardware security keys, [...].

Line 31-32, new: Among the main advantages of PUFs to be used as root of trust in the IoT, [...].

Comment 1.3:

Currently, from the way the paper is written, I cannot clearly see how the response is produced. The authors suggest to provide a high-level figure to assist the understanding of this. This is important and should be provided early on in the paper. (1) How the digital response is produced, (2) a clear statement of which variations this PUF utilizes, and (3) what characteristics this PUF exploits, all need to be CLEARLY stated near the beginning.

Response 1.3:

For a broad and interdisciplinary audience, we decided in our first version of the manuscript to put the details of the PUF mechanism into the supplementary information, whereas the root of variations is discussed in the main manuscript and visualized in Figure 1.

However, to address the concern of reviewer 1 and also to include and answer to a similar comment (comment 2.5.) of reviewer 2, we have added an additional figure (Figure 1h) to clearly show the challenge-response mechanism and within that the digital response (answers point 1) extracted from the inverter output voltages (answers point 3). The variations (answers point 2) are now described in the manuscript in line 93 and are partly shown in Figure 1b when the surface roughness and thickness of the various layers are discussed. Through these extensions we have now created a more comprehensive picture early in the manuscript (section 2.1 of Results) to address all three points of reviewer 1.

The full challenge-response mechanism of the scaled-up system to form a 28-bit including the permutation-based challenge addressing is further described in the supplementary information.

Changes to the manuscript:

Line 93-100, new, (added): In our hybrid PUF approach, we exploit the implicit random variations caused by the material composition, layer thickness and roughness, as well as interface properties between several printed layers as a source of randomness for hardware security. These variations are reflected in the electrical characteristics of our printed EGTs and corresponding inverter structures.

By addressing two inverters (Inv_{ak} , Inv_{bk}) simultaneously and comparing their output voltages, one output bit is generated, based on the voltage difference ΔV_{out} . To enable a comprehensive understanding of the challenge-response mechanism and the corresponding effects that lead to the response bits, we also track the individual inverter output voltages at the comparator input with an analog-to-digital converter (ADC). The full mechanism of the challenge-response generation is shown in Figure 1h. The exact challenge configuration to generate a 28-bit response is further described in the supplementary information.

Line 86, new: Figure 1h:

Figure 1: [...] h) High-level schematic of the challenge and response generation procedure. The inverter pair (Inv_{ak} , Inv_{bk}) addressing is provided by the permutation-based sub-challenge c_k . The comparator output generates the corresponding sub-response r_k , based on the voltage difference ΔV_{out} between Inv_{ak} and Inv_{bk} .

Comment 1.4:

As there are other works related to printed PUF in literature, the authors should provide a qualitative comparison among them (if quantitative comparison is non-trivial). This is to clearly distinguish this work from the others. At moment the clarity here needs improvement.

Response 1.4:

We have included a table comparing qualitatively our work with other printed PUFs (Table R1.4) which is included in the revised supplementary information.

Table R1.4: Comparison between different PE-based PUFs with the hybrid PUF.

PUF	Printing technology	Response generation	PUF type	Response bit-width	PUF metrics (experimental)	Suppl. Ref.	Year
Memory-PUF	Inkjet	Electrical	Weak	1	-	[1]	2018
Resistive CNT PUF	Inkjet	Electrical	Weak	-	-	[2]	2019
SRAM-PUF	Screen	Electrical	Weak	4	-	[3]	2012
Quantum Dot PUF	Inkjet	Optical	Strong	-	-	[4]	2019
This Work (Hybrid PUF)	Inkjet	Electrical	Weak	28	Uniqueness Reliability Bit error Bit aliasing FRR and FAR		2020

From the table and by our revised introduction the conceptual novelty of our work is the realization and comprehensive experimental study including the implementation, fabrication and integration of a printed electronic PUF into an embedded hybrid system.

Our achievements, compared to the related works are:

- extracting physical randomness from thin metal oxide devices in a differential circuit design to form a complex, 28-bit PUF response.
- first experimental access to full security metrics such as uniqueness, reliability, bit aliasing, false acceptance rate and false rejection rate, based on hidden variations.
- statistical analysis based on 15 integrated PUF cores and 3300 measured PUF responses
- the first novel material and manufacturing system that can be compared due to its complexity with silicon PUFs.

Prior existing studies on printed PUFs focused on standalone output voltage measurements of mostly single bit functions. Our approach consequently addresses the design of a hybrid PUF starting from the level of printable materials, the formation of the thin film electrical devices and their circuit behavior up to the embedded system. Only this coherent design, fabrication and characterization approach allows for the understanding and calculation of the security metrics based on a specific challenge-response mechanism. In that context, we believe our analysis could serve as a benchmark for novel material based PUFs to compare their security metrics in future. It should be noted, that our approach requires a high amount of statistical data and reproducibility of the utilized novel materials and technology. Furthermore, with this work we want to give full insights into the environmental effects, influencing PUF operation such as temperature, humidity and noise – as visualized and discussed in the Results section of the manuscript.

In summary, we hope that our results on the hybrid PUF system motivates further research on security devices and systems based on novel materials and provide a promising exploitation path for the design of secure lightweight identification devices in the IoT.

Changes to supplementary information:

Line 29-36, new: We qualitatively compare this work with other state of the art experimentally evaluated PE-based PUFs. To the best of our knowledge no fully verified PE-based PUF including the evaluation of all security metrics has been reported in literature, yet. However, evaluating the PUF security metrics is important to enable a qualitative comparison between different PUF implementations. Nonetheless, currently existing comparable results are listed in Supplementary Table 2. We compare the used printing technology, the type of response generation (electrical or optical), the PUF type (weak or strong PUF), the presented response bit width, as well as experimentally verified PUF security metrics. This includes the uniqueness, reliability, bit aliasing bit errors, false-acceptance rate (FAR), and false-rejection-rate (FRR).

Supplementary Table 2: Qualitative comparison of PE-based PUF implementations. The comparison includes experimentally verified PUFs. The absence of a compared PUF parameter is denoted by a (-).

PUF	Printing technology	Response generation	PUF type	Response bit-width	PUF metrics (experimental)	Suppl. Ref.	Year
Memory-PUF	Inkjet	Electrical	Weak	1	-	[1]	2018
Resistive CNT-PUF	Inkjet	Electrical	Weak	-	-	[2]	2019
SRAM-PUF	Screen	Electrical	Weak	4	-	[3]	2012
Quantum Dot-PUF	Inkjet	Optical	Strong	-	-	[4]	2019
This work (Hybrid PUF)	Inkjet	Electrical	Weak	28	Uniqueness Reliability Bit error Bit aliasing FRR and FAR		2020

Comment 1.5:

Paper(s) on PUFs were recently published in Nature Electronics (within the last two months). Please do a search and cite these in the introduction to assist motivating the importance and directions in the field.

Response 1.5:

In addition to the already cited references of [34, 35, 38, 42, 44, 45, 46], we added the latest (within the last two months) PUF related publications from Nature Electronics (Gao et. al) [18] and Nature Communications (Leem et. al) [40], (Gu et. al) [41]. The work [18] presents a PUF review paper concentrating on general concepts and implementations for silicon PUFs, whereas the works presented in [40], [41] utilize optical inspection for PUF functionality.

Changes to the manuscript:

Line 374, new: [18] Gao, Y., Al-Sarawi, S. F. & Abbott, D. Physical unclonable functions. *Nature Electronics* 3, 81–91 (2020).

Line 418, new: [40] Leem, J. W. et al. Edible unclonable functions. *Nature communications* 11, 1–11 (2020).

Line 419-420, new: [41] Gu, Y. et al. Gap-enhanced Raman tags for physically unclonable anticounterfeiting labels. *Nature communications* 11, 430 1–13 (2020).

Comment 1.6:

The math is wrongly formatted. Only single letter variables should be in italic. Other items such as: max, amp, RH, comp, out, core, in, BE, BA, HD...etc should have italic removed. If for example you put RH in italic it means R x H. Use this visualisation to correct all; the math.

Response 1.6:

Thanks for pointing it out, we corrected the math formatting in the manuscript accordingly.

Comment 1.7:

Whilst it is correct that there is always a space between a number and the physical units, the deg Centigrade symbol is an exception. Remove the space here.

Response 1.7:

We adjusted the '°C' formatting in the manuscript accordingly.

Comment 1.8: Please carefully check the references. There are many errors. For example conventions concerning capitalisation are wrong. All journal names should be in title case, not sentence case. All acronyms should be capitalised.

Response 1.8:

Thanks for highlighting this error source, we revised the reference style matching the Nature reference style.

Comment 1.9:

On page 2 you have "Si-PUFs can be further classified...." In English, it is verboten to begin a sentence with an acronym. All sentences must begin with a real English word.

Response 1.9:

Thanks for highlighting this error, we use the full word now.

Changes to the manuscript:

Line 37, old: Si-PUFs can be further classified [...].

Line 37, new: Silicon-based PUFs can be classified [...]

Reviewer #2:

This is a very well written paper with very interesting and well documented results in terms of creating a printed PUF circuit that can serve as a unique intrinsic identifier. The methods and the results are described in great detail that would allow another to replicate the experiment. The material is a definite interest to the research community.

There are several areas where this paper can be improved:

Comment 2.1:

Page 1 line 15 "high entropy". The paper claims a 28 bit PUF. But there are 8 inverters. The produced bits have 0.45 bias with similar bits on other devices. I guess high entropy is a relative term. You might want to elaborate.

Response 2.1:

We fully agree that the term 'high entropy' is not well defined in this context and needs clarification. In general, in the context of information theory, entropy relates to the uncertainty of an outcome and is interlinked with randomness. This general definition is in principle applicable for all kinds of uncertainties, as e.g. in probability theory, chaos theory, statistics, cryptography, to name but a few. Due to the nature of additive printing processes, ink substrate interactions, as well as intrinsic material properties based on thin film morphology and interface effects, PE technology underlies higher variation compared to silicon-based electronics. This leads to greater uncertainty in term of unpredictability in the circuit characteristics. For that reason, we talk about high entropy in terms of random/uncontrollable variations. In summary, this discussion mainly refers to the physical entropy source.

The NIST specification 800-90 recommends determining the so-called min-entropy to assess the worst-case information entropy of a binary source. The min-entropy metric is based on the distribution of 0s and 1s. Thereby, the probability of having a 0 is denoted by p_0 , whereas the probability of a 1 is p_1 . In general, the min-entropy H_{\min} is calculated according to Equation (1):

$$H_{\min} = -\log_2(p_{\max}) \quad (1)$$

where $p_{\max} = \max(p_0, p_1)$.

In the context of PUFs, the min-entropy metric considers the bias of the PUF responses. As shown in Figure 4a in the manuscript, the average bit aliasing, which corresponds to the bias, is $\mu_m = 44.5\%$ for the fabricated and evaluated hybrid PUFs. This leads to a min-entropy of $H_{\min} = -\log_2(0.555) = 0.849$, which denotes that the actual information content of a 28-bit response is limited to ≈ 23.772 bits. However, for our simulation results we determined a mean bit aliasing value of $\mu_m = 49.8\%$, which leads to a min-entropy value of $H_{\min} = -\log_2(0.502) = 0.994$. This value indicates the theoretical performance that can be expected for greater sample sizes in our approach.

Finally, we want to summarize that there are various interpretations of the term 'entropy'. As suggested by the reviewer, we will now distinguish between the entropy source (which refers to the variations and materials properties) and the information entropy of the binary PUF responses.

To reduce possible misunderstandings, we changed the wording in the abstract accordingly. Furthermore, we include the calculations as shown above, in the supplementary information and discuss this topic.

Changes in the manuscript:

Line 15, old: The security primitive is high-entropy, [...].

Line 15, new: The security primitive provides high intrinsic variation, [...].

Changes to the supplementary information:

Line 65 ff., new: 1.5.1 Entropy of the PUF responses

To determine the entropy of random numbers the min-entropy estimation is widely used, as recommended in the NIST specification 800-90. The min-entropy metric is based on the distribution of 0s and 1s. Thereby, the probability of having a 0 is denoted by p_0 , whereas the probability of a 1 is p_1 . In general, the min-entropy H_{\min} is calculated according to Equation (1):

$$H_{\min} = -\log_2(p_{\max}) \quad (1),$$

where $p_{\max} = \max(p_0, p_1)$.

In the context of PUFs the min-entropy metric considers the bias of the PUF responses. As shown in Figure 4a in the manuscript, the average bit aliasing, which corresponds to the bias, is $\mu_m = 44.5\%$ for the fabricated and evaluated hybrid PUFs. This leads to a min-entropy of $H_{\min} = -\log_2(0.555) = 0.849$, which denotes that the actual information content of a 28-bit response is limited to ≈ 23.772 bits. However, for our simulation results we determined a mean bit aliasing value of $\mu_m = 49.8\%$, which leads to a min-entropy value of $H_{\min} = -\log_2(0.502) = 0.994$. This value indicates the theoretical performance that can be expected for greater sample sizes in our approach.

Comment 2.2:

Page 1 lines 21 to 29: you are not distinguishing between the true source of randomness vs. something that uses that to produce pseudorandomness (e.g., a cryptographic algorithm). PUF you are describing is in the former category. The "28 bit" PUF does not provide sufficient entropy into the cryptographic algorithms given that there are 8 inverters and 0.45 bias across devices.

Response 2.2:

As pointed out by the reviewer and explained in Response 2.1, the PUF responses in their current form and the amount of statistical data does not hold a high enough entropy for cryptographic applications. As noted in lines 30,31 of the manuscript, the design goal of the hybrid PUF was to investigate PUFs for secure, lightweight identification for IoT devices.

In general, PUFs use a natural entropy source that is expected to be random to generate PUF responses. Therefore, true random numbers can be generated by PUFs. However, most physical realization of PUFs suffer from systematic errors and noise, making it difficult to enable true random responses without post-processing – which provides another unwanted attack vector.

The experimentally demonstrated bias ($\mu_m = 44.5\%$) based on 15 PUF cores for the hybrid PUF and a theoretical bias of $\mu_s = 49.8\%$, based on python simulations are already promising and in principle the circuit complexity and architecture including the addressing and readout circuits could be optimized for cryptographic applications.

Changes in the manuscript:

Line 190, new: The experimentally obtained a bit aliasing of $\mu_m = 44.5\%$ indicates a bias of the PUF responses towards logic '0'. However, the simulated theoretical bit aliasing for the presented hybrid PUF is $\mu_s = 49.8\%$. This shows that the bit aliasing can be improved to provide a close to true random bit sequence, suitable for cryptographic applications with the presented approach. The hybrid PUF's response entropy and its capabilities regarding identification are discussed in the supplementary information.

Comments 2.3:

Page 1 line 32. "PUFs used as hardware security keys". PUFs can be used to derive the keys. A function is not a key. Also, the 22% error in your PUF would preclude getting a bit stable key unless lots of post processing is done (and that would be difficult with printed electronics).

Response 2.3: Thank you for pointing it out, we have refined our language and corrected the sentence accordingly. We agree that reducing the error rates by increasing the reliability is a

crucial point to be addressed in future work on PE-based PUFs. This can be achieved with temperature compensated low-overhead architectures, passivation and encapsulation.

Changes in the manuscript:

Line 249-251, new: Regarding future work in the area of printed PUFs several points need to be further investigated. As shown in our results, temperature stability needs to be improved to increase reliability and to enable bit-stable PUF responses.

Comment 2.4:

Page 1 lines 28-29. Loftstrom paper (24) is not a PUF. It does not define its circuit as having an input and therefore is not a function. PUFs came after Gassend 2002 paper which requires input (challenge) and an output with output determined by manufacturing variation. Butterfly PUF (22) is a bistable PUF, not delay PUF. The first delay PUF paper is arguably the Gassend CCS 2002 paper.

Response 2.4:

We agree that the challenge-response mechanism, that defines a PUF and its functionality, is not declared in the Lofstrom paper. The work of Gassend et. al [14] shows and discusses the challenge-response functionality for the first time. We therefore decided to replace the reference of the Lofstrom paper [25], by related work on analog-PUFs by Yang et. al) [24]. We also updated the references for the Butterfly PUF, which is classified as a bistable PUF.

Changes in the manuscript:

Line 372-373, old: [25] Lofstrom, K., Daasch, W. R. & Taylor, D. *Ic identification circuit using device mismatch*. In Solid-State Circuits Conference, 2000. Digest of Technical Papers. ISSCC. 2000 IEEE International, 372–373 (IEEE, 2000).

Replaced by:

Line New: 386-387, new: [24] Yang, K., Dong, Q., Blaauw, D. & Sylvester, D. *8.3 A 553F² 2-transistor amplifier-based physically unclonable function (PUF) with 1.67% native instability* in 2017 IEEE International Solid-State Circuits Conference (ISSCC) (2017), 146–147.

Comment 2.5:

Page 4 - 5 lines 122 to 127. It is unclear how the 8 inverters are expanded to 28-bit challenge using permutations. Might want to describe that more. Also elaborate whether the bits generated is true entropy or pseudo-entropy.

Response 2.5:

To show the generation of the 28-bit response-challenge mechanism, based on 8 inverters, we include an additional figure (Figure 1h) and further show the permutation sequence of a challenge. The inverter input and outputs are readdressed in a lexicographic order, in order to derive a 28-bit response. Therefore, each inverter address will occur $M-1$ times in the response with $M=8$ inverters. Furthermore, the challenge-response mechanism is described in more detail in the supplementary information.

As elaborated in our response to comment 2.1 we have clarified the terms random entropy source, have shown the potential of creating a min entropy of 1 and elaborated on the current experimental results of the hybrid PUF system. The corresponding changes in the manuscript have been listed above (comment 2.1).

Since true random numbers are not biased towards logic '0' or '1', the results obtained from our statistical analysis on the hybrid PUF show that the PUF responses are pseudo-random. However, most computer-generated random numbers are generated using (computational) deterministic cryptographic algorithms. In practice, pseudorandom numbers can be enough, also for security-critical applications.

At this point we want to note that these issues exist for most of the PUF designs and is not limited to the hybrid PUF.

Changes in the manuscript:

Line 86, new: Figure 1h:

Figure 1: [...] h) High-level schematic of the challenge and response generation procedure. The inverter pair (Inv_{ak} , Inv_{bk}) addressing is provided by the permutation-based sub-challenge c_k . The comparator output generates the corresponding sub-response r_k , based on the voltage difference ΔV_{out} between Inv_{ak} and Inv_{bk} .

Comment 2.6: Page 6 line 177. It seems like μ_m of 44.5% represent a min entropy reduction of $-\log(.55)$ reducing 8 bit entropy to 6.8 bits. This assumes adversary can use other like devices to break another via across device bit correlation.

Response 2.6: We are not sure, we interpret this comment right. We agree that a μ_m of 44.5 % represents a non-ideal value and could result in a security threat. We have included detailed calculations of the min-entropy in our response of comment 2.1, as well as in the revised manuscript in the supplement material.

Comment 2.7:

Page 8 line 226. A reliability of 78.5%, meaning error rate > 20% is actually quite high in the silicon PUF world. For identification, this would require an even longer response length for reliable authentication (for a given false positive and false negative goal).

Response 2.7: We fully agree, that the reliability at this point is, in comparison to state-of-the art, silicon-based PUFs relatively low. This stems from various effects, as in current Si-based PUF implementations, stabilization methods for high reliability, uniqueness and bit-errors are deployed, such as temporal majority voting. However, this post-processing doesn't come without a cost, as Helper-Data is required, that stores information for stable PUF response generation. This provides an attackable weak spot in hardware-based security. The presented reliability of the hybrid PUF consists of raw PUF responses only. The reliability values are comparable, if even better, than first Si-based PUF implementations. One should note, that our hybrid PUF presents the first fully evaluated device utilizing emerging technologies such as inkjet-printing and metal oxide materials. However, we agree, that the reliability could be improved in the future. Possible solutions would include proper passivation, encapsulation and research in low-overhead temperature stable PE-based architectures. We therefore added a discussion on improving the reliability and highlight future research perspectives, regarding inkjet-printed PUFs and their materials. To the best of our knowledge, our work shows the first realization of a PE-based PUF with comprehensive statistics on security metrics based on experimental data. For the time being it is not possible to compare the reliability results with other related works not based on silicon, due to the novelty of our work.

To determine the identification capabilities of the hybrid PUFs, we determine the fuzziness of the PUF responses. Therefore, we compute plot the intra-HD and inter-HD distributions based on experimental and simulation data. Figures R2.7 (a) and R2.7 (b) show the corresponding distributions. The area enclosed between both curves determines the fuzziness of the responses. The area can be split by the identification threshold (th_{id}) into the left region, which is also referred to as the false-acceptance-rate (FAR) and the right region, the so-called false-rejection-rate (FRR). In the ideal case and if the enclosed area is zero, all PUF responses can be distinguished without errors. However, the ideal case is never reached without post-processing since fabricated raw PUFs underlie variations induced through the imperfect manufacturing process as well as changing operating conditions.

Figure R2.7: Intra-HD and inter-HD distributions on the basis of (a) experimental data, and (b) simulation data.

The FAR and FRR values are calculated according to Equation (R2.7-1) and Equation (R2.7-2), respectively:

$$FAR(th_{id}) = \int_0^{th_{id}} P_{inter}(x) dx \quad (R2.7-1)$$

$$FRR(th_{id}) = \int_{th_{id}}^{L_{max}} P_{intra}(x) dx \quad (R2.7-2)$$

where $P_{inter}(\cdot)$ and $P_{intra}(\cdot)$ are the probability density functions of the inter-HD and intra-HD distributions. Since the FAR and FRR values are typically very small numbers, it is common practice to use the $\log_{10}(\cdot)$ representation. Typical FAR and FRR values used in identification systems can reach from -3 up to -12 (after post-processing).

Basically, the FAR and FRR values depend on the selected identification threshold value. Two often used approaches to set the identification threshold is (1) to use the intersection point between both distributions and (2) use the so-called equal-error-rate (EER) where FAR=FRR.

Firstly, we use the experimental response data and calculate the FAR and FRR values for the former identification threshold. The resulting values are FAR=-2.23 and FRR=-1.71. Furthermore, we compute the values for the EER, which result in FAR=FRR=-1.83. To the best of our knowledge, this is the first assessment of the identification capabilities of a PE-based PUF. Even in the more matured research field of silicon-based PUFs, such detailed statistical evaluations are rare. At this point we want to note that our evaluations are based on raw PUF responses without additional post-processing, such as error-correction.

For our simulation data, the resulting values are FAR=-2.21 and FRR=-2.68. For the equal-error-rate (EER) the values are FAR=FRR=-2.32. The results show that the experimental results are in good agreement with our simulations. Additional post-processing could further improve the identification capabilities of the hybrid PUF and could be tackled in future work.

Changes in the supplementary information:

Line 77, new: 1.6 Hybrid PUF identification

To investigate the identification capabilities of the hybrid PUF, we perform evaluations based on the intra-HD and inter-HD distributions. The intra-HD is a measure of the reproducibility of the PUF responses for a fixed challenge and under the impact of changing operating conditions. The inter-HD indicates the uniqueness of the responses generated by different PUFs. Supplementary Figure 4a shows the intra- and inter-HD distributions for our measured PUF responses under humidity and voltage variations. The solid black line shows the intra-HD Gaussian distribution, whereas the dash-dot red line shows the inter-HD Gaussian distribution, respectively. The enclosed area below both lines divides into the two regions left and right from the intersection value. The left region is denoted as the false-acceptance rate (FAR), whereas the right one is the false-rejection rate (FRR). To ensure a proper identification, both the FAR and FRR should be minimized and the PUF responses should contain enough entropy with respect to the sample size. As there is an overlap between both, the intra- and the inter-HD variation distributions, some PUFs cannot be distinguished without additional post-processing.

Supplementary Figure 4: Intra- and inter-hamming distance (HD) distributions. (a) Gaussian distributions based on our measured data. (b) Gaussian distributions based on simulated data.

The x-value of the intersection is the ideal threshold value to distinguish between PUF responses when applying a binning technique. If the HD between the database entry and the measured response is less than this threshold, the identification is successful. If there is no overlap between both distributions, the identification can be assumed errorless, if the threshold is placed somewhere in between. Supplementary Figure 4b shows the corresponding intra- and inter-HD Gaussian distributions for the simulated data of 150 printed PUF cores. The simulation data used has been generated in our prior work and refers to the worst-case considerations where a noise level of 10 mV is applied. The standard deviation of the inter-HD can be expected to decrease for larger sample sizes. The overlap in the plot between both distributions is small, which implies a low identification error of the hybrid PUFs. A three-bit error correction reduces the intra-HD to zero and therewith also eliminates the overlapping area. To further assess the identification capabilities based on the raw PUF responses (without additional error correction), we compute the FAR and FRR values based on experimental and simulation data. The FAR and FRR values are calculated according to Equation (2) and Equation (3), respectively:

$$FAR = \int_0^{th_{id}} P_{inter}(x) dx \quad (2)$$

$$FRR = \int_{th_{id}}^L P_{intra}(x) dx \quad (3)$$

where $P_{inter}(\cdot)$ and $P_{intra}(\cdot)$ are the probability density functions of the inter-HD and intra-HD distributions. Since the FAR and FRR values are typically very small numbers, it is common practice to use the $\log_{10}(\cdot)$ representation. Typical FAR and FRR values used in identification

systems reach from -3 up to -12 (after post-processing). Basically, the FAR and FRR values depend on the selected identification threshold value. Two often used approaches to set the identification threshold is (1) to use the intersection point between both distributions and (2) use the so-called equal-error-rate (EER) where $FAR=FRR$. We use the experimental response data and calculate the FAR and FRR values for the former identification threshold. The resulting values are $FAR=-2.23$ and $FRR=-1.71$. Furthermore, we compute the values for the EER, which result in $FAR=FRR=-1.83$. To the best of our knowledge, this is the first assessment of the identification capabilities of a PE-based PUF. Even in the more matured research field of silicon-based PUFs, such detailed statistical evaluations are rare. At this point we want to note that our evaluations are based on raw PUF responses without additional post-processing, such as error-correction. For our simulation data, the resulting values are $FAR=-2.21$ and $FRR=-2.68$. For the EER the values are $FAR=FRR=-2.32$. The results show that the experimental results are in good agreement with our simulations. However, additional post-processing could further improve the identification capabilities of the hybrid PUF.

Comment 2.8: Page 8 line 234 "root of trust" usually refers to a bit-stable key. An error rate of 20%+ is difficult to address even with error correction. And on a printed circuit I am not sure how that's going to be addressed. There is a lot of future work here.

Response 2.8:

Concerning the quantitative analysis of the error rates, please refer to our response to your comment 2.7. In addition, we would like to point out, that from an information security point of view, the "root of trust" describes a source that can always be trusted. Typically, in cryptographic systems the security is dependent on binary keys to encrypt or decrypt data, generate digital signatures, to name but a few. The root of trust is inaccessible from the outside and guarantees the authenticity of the overall system. Most commonly, the root of trust comprises functionalities for random number generation, key derivation, secure memory etc. Device-unique keys are often injected into the system during the manufacturing process by an external party, which depicts a potential security threat. However, the advantage of injected keys is their perfect reproducibility. On the other hand, PUFs are an option to derive binary keys from an inherent entropy source. In the context of root of trust, PUFs can provide a higher security level, since no external party is needed to inject device-unique keys. Furthermore, in PUFs use an intrinsic storage (variations), which mitigates the threats emanating from memory leakage attacks.

PE technology, particularly the unique property of decentralized manufacturing, allows to fabricate circuits at the manufacturing site and not necessarily at a foundry. In this context, PE-based PUFs can leverage from trusted supply chains, which further enhances the security level of the keys. Consequently, PE-based PUFs can enhance the overall security level of a root of trust by covering secure key derivation and trusted manufacturing.

To address the reviewer's comment regarding possible future work we refer to response 2.3 and the corresponding changes in the manuscript.

Comment 2.9:

Page 10 lines 316 to 326. Why use w and t , two different variables. These two eqns look almost the same. Isn't one just 100 minus the other?

Response 2.9:

Equation (5) in the manuscript defines how to calculate the reliability metric. The reliability metric determines how reproducible PUF responses are under the impact of changing operating conditions. For our statistical evaluations, we use T different operating conditions, as shown in Supplementary Table 1 in the Supplementary Information.

On the other hand, we use Equation (4) from the manuscript to compute the bit errors occurring when regenerating a PUF responses multiple times while using the same challenge. In this case, W determines how often the PUF response is regenerated.

We hope this clarifies the difference between the equations (4) and (5).

Comment 2.10:

Page 11 line 360. I don't see an italicized publication name.

Response 2.10:

We corrected the citation style of mentioned publication accordingly.

We hope our answers could clarify the reviewer questions and look forward to a positive response.

With best regards,
Jasmin Aghassi-Hagmann

REVIEWERS' COMMENTS:

Reviewer #1 (Remarks to the Author):

I am happy that the authors addressed all my comments.

Before uploading the final files for publication please carefully check the references again. For examples, anything in italic, in the references, should be in title case not sentence case. This has still not been fully corrected.

Reviewer #2 (Remarks to the Author):

Thank you for the thoughtful comments and edits to address the reviewers' concerns. The revised manuscript is even better.

Point to consider for future work: I am not sure whether the randomness / entropy measurements you are taking assumes that the adversary doesn't know the challenge schedule. In otherwords, a part of the bit aliasing and bias effects you are measuring may be an artifact of the randomness of the challenge schedule itself, which can be reversed if the adversary know the schedule design.

In any case, very nice work!

Point-by-point response letter to the reviewers:

We would like to thank the reviewers for their valuable feedback on our manuscript. Please find our detailed comments and changes in the manuscript below.

Reviewer #1:

I am happy that the authors addressed all my comments.

Comment 1:

Before uploading the final files for publication please carefully check the references again. For examples, anything in italic, in the references, should be in title case not sentence case. This has still not been fully corrected.

Response 1:

Thank you for pointing this out. We have corrected the references list to comply with the style guidelines provided by the Nature Publishing Group.

Reviewer #2:

Thank you for the thoughtful comments and edits to address the reviewers' concerns. The revised manuscript is even better. In any case, very nice work!

Comment 2:

Point to consider for future work: I am not sure whether the randomness / entropy measurements you are taking assumes that the adversary doesn't know the challenge schedule. In otherwords, a part of the bit aliasing and bias effects you are measuring may be an artifact of the randomness of the challenge schedule itself, which can be reversed if the adversary know the schedule design.

Response 2:

We fully agree that the challenge schedule can affect the randomness and entropy measurements in a negative manner. For that reason, we have created challenges causing that all PUF core inverters are addressed equally often to prevent from such negative influences. Thus, our statistical results show, that the experimentally determined PUF responses are slightly biased towards logic '0'. In our future work, we will focus on introducing techniques to further strengthen the attack resilience of the hybrid PUF.

We hope our answers could clarify the reviewer comments and look forward to a positive response.

With best regards,
Jasmin Aghassi-Hagmann